# Prognostic Role of Androgen Receptor Expression in HER2+ Breast Carcinoma Subtypes

**DOI:** 10.3390/biomedicines10010164

**Published:** 2022-01-13

**Authors:** Sandra Orrù, Emanuele Pascariello, Giovanni Sotgiu, Daniela Piras, Laura Saderi, Maria Rosaria Muroni, Ciriaco Carru, Caterina Arru, Cristina Mocci, Giampietro Pinna, Raffaele Barbara, Paolo Cossu-Rocca, Maria Rosaria De Miglio

**Affiliations:** 1Department of Pathology, “A. Businco” Oncologic Hospital, ARNAS Brotzu, Via Edward Jenner 1, 09121 Cagliari, Italy; sandra.orru@aob.it (S.O.); anatomiapatologica.businco@aob.it (E.P.); cristina.mocci@aob.it (C.M.); g.pinna33@studenti.unica.it (G.P.); 2Department of Medical, Surgical and Experimental Sciences, University of Sassari, Via P. Manzella 4, 07100 Sassari, Italy; gsotgiu@uniss.it (G.S.); lsaderi@uniss.it (L.S.); mrmuroni@uniss.it (M.R.M.); rocco@uniss.it (P.C.-R.); 3Struttura Complessa Epidemiologia e Registro Tumori Nord Sardegna, ATS Sardegna, Via Rizzeddu 2, 07100 Sassari, Italy; dani.piras@atssardegna.it; 4Department of Biomedical Sciences, University of Sassari, Viale San Pietro 43, 07100 Sassari, Italy; carru@uniss.it (C.C.); 30039590@studenti.uniss.it (C.A.); 5Department of Radiotherapy, “A. Businco” Oncologic Hospital, ARNAS Brotzu, Via Edward Jenner 1, 09121 Cagliari, Italy; raffaele.barbara@aob.it; 6Department of Diagnostic Services, “Giovanni Paolo II” Hospital, ASSL Olbia-ATS Sardegna, Via Bazzoni—Sircana, 07026 Olbia, Italy

**Keywords:** HER2 expression, AR expression, breast cancer, prognosis, immunohistochemistry

## Abstract

HER2+ breast cancer (BC) is an aggressive subtype representing a genetically and biologically heterogeneous group of tumors resulting in variable prognosis and treatment response to HER2-targeted therapies according to estrogen (ER) and progesterone receptor (PR) expression. The relationship with androgen receptors (AR), a member of the steroid hormone’s family, is unwell known in BC. The present study aims to evaluate the prognostic impact of AR expression in HER2+ BC subtypes. A total of 695 BCs were selected and reviewed, AR, ER, PR and HER2 expression in tumor cells were examined by immunohistochemical method, and the SISH method was used in case of HER2 with equivocal immunohistochemical score (2+). A high prevalence of AR expression (91.5%) in BC HER+ was observed, with minimal differences between luminal and non-luminal tumor. According to steroid receptor expression, tumors were classified in four subgroups, including BC luminal and non-luminal HER2+ expressing or not AR. The luminal BC HER2 + AR+ was associated with lower histological grade, lower tumor size, higher PR expression and lower HER2 intensity of expression (2+). Also, the non-luminal tumors AR+ showed lower tumor size and lower prognostic stage but frequently higher grade and higher HER2 intensity of expression (3+). These findings should suggest a different progression of luminal and non-luminal tumors, both expressing AR, and allow us to speculate that the molecular mechanisms of AR, involved in the biology of BC HER2 + AR+, differ in relation to ER and PR expression. Moreover, AR expression may be a useful predictor of prognosis for overall survival (OS) in HER2+ BC subtypes. Our findings suggest that AR expression evaluation in clinical practice could be utilized in clinical oncology to establish different aggressiveness in BC HER2+ subtypes.

## 1. Introduction

Breast cancer (BC) is a heterogeneous disease enclosing several entities with different morphologic, prognostic and therapeutic features [1]. Invasive breast cancer is classified according to histology and immunohistochemistry (i.e., ER, PR, HER2 overexpression, and/or HER2 gene amplification, and Ki67 proliferation index) [1,2]. Surrogate molecular classification of BC by means of immunohistochemistry defines specific subtypes, such as Luminal A (ER and PR positive, HER2 negative, low Ki67), Luminal B (ER positive, PR positive/negative, HER2 negative, high Ki67), Luminal B HER2+ (ER positive, PR positive/negative, HER2 positive, any Ki67), HER2+ non-luminal (ER-PR negative, HER2 positive, high Ki67), and triple negative (ER-PR-HER2 negative, high Ki67) [3,4]. BC immunohistochemistry surrogate classification has an actual utility in the management of BC patients having significant prognostic and predictive value [4].

HER2+ BCs include different histological subtypes: invasive BC no special type (IBC-NST), lobular, micropapillary, apocrine, rarely mucinous [5]. Immunophenotypically, HER2+ BCs variably express hormone receptors.

HER2 protein overexpression or gene amplification, which accounts for ~15–18% of all BC, is frequently associated with high invasiveness and worse prognosis without appropriate therapy [6]. HER2 signaling activates PI3K/AKT and RAS/RAF/MAPK pathways, favoring cell proliferation, growth, invasion, and angiogenesis [7]. Sequential chemotherapy combined with anti-HER2 therapy is the landmark for HER2+ BC treatment, both in the neoadjuvant, adjuvant and metastatic setting [8]. Five drugs were made available by the FDA-U.S. for HER2+ BCs treatment: trastuzumab, lapatinib, neratinib, TDM-1, and pertuzumab [9]. Not all HER2+ BC patients clinically benefit from trastuzumab treatment owing to intrinsic or acquired resistance [10], which depends on persistence of HER2 activation, associated with other EGFR members compensatory activations, or structural aberrations of HER2 protein altering trastuzumab binding, activation/inactivation of members of PI3K/AKT downstream to HER2, or activation of different tyrosine-kinases receptors [11,12,13,14,15].

The androgen receptor is a member of the steroid nuclear receptor superfamily, and it is widely expressed in different subtypes of breast cancer, including HER2+ tumors. It could be a promising prognostic and predictive factor and therapeutic target in BC [16,17,18,19]. AR expression and its biologic effects can vary, depending on ER expression [20,21,22]: ~90% of ER-positive BCs express AR, which inhibits ER activity and improves prognosis, though it can induce resistance to tamoxifen and/or aromatase inhibitors, which can be reverted by AR inhibitors [23,24,25,26]. AR was found in ~20% of Triple Negative BCs, with controversial findings on its prognostic role [27,28]. Most HER2+ BCs overexpress AR [20,29], but its prognostic role is unclear [30,31]. Several studies showed that AR can promote the growth of HER2 +  BC cells by cross-talking with the HER2 signaling. Ni et al. found that AR activated by androgen/WNT7B binds to FOXA1 and β-catenin and triggers HER2 and HER3 downstream, promoting the proliferation [32]. Lin et al. showed that co-expression of HER2, ER, and AR reduces tumor invasiveness, improving patient outcomes [33]. The expression of AR in HER2+, ER-negative BC patients has been associated with worse clinical outcomes [23]. The AR receptor blockade could represent a therapeutic option in HER2+ BC. On the other hand, HER2+ BC patients treated with first-line trastuzumab show better disease-free survival (DFS) and overall survival (OS) in case of high AR expression [34].

The aim of the present study was to evaluate AR expression in a large series of HER2+ BC, to establish its prognostic role and clinical-pathological relationship.

## 2. Materials and Methods

A retrospective cohort of 695 BC patients, diagnosed between 2012 and 2021, was recruited through a complete review of surgical samples and medical records from the archives of the Department of Histopathology of Oncologic Hospital in Cagliari, Italy. The inclusion criteria were histologically invasive BC, HER2 positive expression determined by immunohistochemistry (IHC) and/or and in situ hybridization (SISH) assay, availability of formalin-fixed, paraffin embedded (FFPE) tumor blocks and clinico-pathological data. The exclusion criteria were in situ BC and HER2 negative status. Three µm-thick tissue sections of FFPE specimens were cut for hematoxylin and eosin staining (H&E), immunohistochemistry, and SISH analysis.

The study protocol was approved by the local research ethics committee (File number PG/2021/14264); and followed the Italian law on guidelines for the implementation of retrospective observational studies (G.U. n. 76, 31 March 2008). Only coded data were collected in order to protect patient confidentiality.

### 2.1. Immunohistochemistry

The immunohistochemistry analysis was performed using specific antibodies against ER, Clone SP1 (Ventana Medical Systems Inc., Tucson, AZ, USA); PR Clone 1E2 (Ventana Medical Systems); Ki67, Clone 30-9 (Roche Diagnostics K.K., Tokyo, Japan); AR Clone SP107 (Cell-MarqueTM, Rocklin, CA, USA); HER2 PATHWAY Clone 4B5 (Ventana Medical Systems). Immunostaining was performed using the Ventana Benchmark XT staining system with Optiview DAB detection kit. In cases of HER2 with equivocal immunohistochemical score (2+), we performed HER2 gene amplification by ultra-View SISH Detection Kit (Ventana Medical Systems). Evaluation of immunostaining and SISH for HER2 was based on American Society of Clinical Oncology/College of American Pathologists (ASCO/CAP) recommendations [6]. ER, PR and AR expression was considered positive if at least 10% immunostained tumor nuclei were detected in the sample [17]. Ki67 was scored low if <14% of tumor nuclei were positive and high if ≥14% of tumor nuclei were positive [35].

### 2.2. Baseline Data

Demographic and clinico-pathological information were extracted from medical records, as well as age at diagnosis, histologic type, histologic grade, tumor site, and TNM classification. Histologic tumor type was established according to the UICC-WHO criteria [5]. Histologic tumor grade was determined in accordance with the Nottingham guideline [36]. TNM staging was described in accordance with the 8th edition of the American Joint Committee on Cancer criteria (AJCC) [37]. Lymph node ratio was described as the ratio between the positive lymph nodes number and the evaluated lymph nodes number. The cut-off points of lymph node ratio were: <0.21, 0.21–0.65, and >0.65 [38]. The time between the date of BC diagnosis and the date of death define the OS.

### 2.3. Statistical Analysis

Quantitative variables were described by median and interquartile range (IQR) according to non–normal distribution of variables, whereas absolute and relative (percentages) frequencies were used for qualitative variables. Statistical differences for qualitative variables were evaluated using Chi2 or Fisher’s exact tests, when appropriate. Logistic regression analysis was performed to evaluate the association between mortality, IHC BC subtypes, clinico-pathological features, and molecular variables. A Kaplan–Meier curve and Log-Rank test were performed to describe OS according to IHC BC subtypes, clinico-pathological and molecular variables. The statistical significance was set up at *p* < 0.05. Statistical analysis was carried out using STATA^®^16 (StataCorp, College Station, TX, USA).

## 3. Results

A total of 695 HER2+ BC patients were selected. Tumors were grouped into four subtypes based on hormone receptors (ER, PR and AR): Luminal B (LB) HER2 + AR+ (ER ≥ 10%, PR ≥ 10% or < 10%, and AR ≥ 10%); Luminal B HER2 + AR− (ER ≥ 10%, PR ≥ 10% or < 10%, and AR < 10%); HER2 + AR+ (ER < 10%, PR < 10% and AR ≥ 10%); HER2 + AR− (ER < 10%, PR < 10%, and AR < 10%). Table 1 shows subtypes proportion based on the state of the hormonal receptors and the results of HER2 intensity immunostaining for each subtype. Figure 1 shows representative IHC positive results obtained for AR and HER2, and SISH positive results for HER2 with equivocal immunostaining intensity (2+). All tumors with HER2 IHC 2+ included in the study were SISH positive.

The median (range) age at diagnosis was 58 (27–91) years, with 363 (52.2%) equal to or older than 58 years. 590 (85.0%) tumors were ductal, 30 (4.3%) lobular, and 39 (5.6%) apocrine; less common were other histologic types, such as mixed (ductal + lobular, 1.9%), micropapillary/papillary (2.2%), and mucinous (1.0%). Pathological tumor staging was pT1 in 203 (32.6%) cases, pT2 in 288 cases (46.2%); pT3 in 27 cases (4.3%); and pT4 in 52 cases (8.4%). Pathological lymph node status was distributed as follows: pN0 in 321 (55.9%); pN1 in 134 (23.3%); pN2 in 73 (12.7%); pN3 in 46 (8.0%). A total of 54 (7.8%) were metastatic; 83.7% of HER2-positive BC were G3. Ninety nine out of 695 patients (14.2%) died (Table 2).

AR positivity was detected in 91.5% of all tumors, specifically in 95.2% of luminal and in 84.2% of non-luminal subtypes.

The median age at diagnosis of HER2+ AR+ BC patients were 59 years (*p*: 0.045) and showed the following features: histological grade 2 was found in 17.3% of AR+ tumors vs. 5.2% of AR− tumors (*p*: 0.020), pT1 in 33.7% of AR+ tumors vs. 20.8% of AR− ones (*p*: 0.020), prognostic stage I in 42.8% AR+ BC vs. 16.3% of those with AR negativity (*p*: 0.001). Moreover, HER2+ AR+ BCs were more frequently associated with ER and PR expression ≥ 10% (*p* < 0.0001 and *p* < 0.001, respectively). There were no significant differences between site, histological type, lymph node status and ratio, metastasis, ki67 and HER2 expression (Table 3).

Luminal and non-luminal HER2+ AR+ BCs showed a higher age at diagnosis compared to luminal and non-luminal HER2+ AR− BCs. HER2+ AR− and HER2+ AR+ subtypes showed less frequently a lobular histotype compared to luminal HER2+ AR− and AR+ subtypes. Mixed histological type was most represented in LB HER2+ subtypes expressing AR; HER2+ AR+ subtype was mainly associated with apocrine phenotype, while HER2+ AR− showed mucinous phenotype more frequently compared to the other subtypes. Histological grade G2 was mainly found in LB HER2+ AR+ (25.5%). HER2+ subtypes without AR expression showed the highest pT3 and pT4 rates, accounting for 13.5% and 10.8% (HER2+ AR−), and 12.5% and 12.5% (LB HER2+ AR−), respectively.

Prognostic stage I was found more frequently in LB HER2+ AR+ (55.5%), and less frequently in HER2+ AR− (6.3%). On the other hand, stage III was mainly found in HER2+, both AR− and AR+ (28.1% vs. 28.8). HER2+ AR− and HER2+ AR+ subtypes showed a significantly higher frequency of HER2 expression with score 3+ (Table 4).

### Prognostic Indicators according to HER2+ BC Subtypes

Factors influencing survival in the univariate analysis were: LB HER2+ AR+ subtype (*p*: 0.010); age at diagnosis (*p* < 0.0001); histological type (*p*: 0.006); histological grade (*p*: 0.003); tumor size (*p* < 0.0001); lymph node status and ratio (*p* < 0.0001 for both); prognostic stage (*p* < 0.0001); metastasis (*p* < 0.0001); Ki-67 expression (*p*: 0.002); ER, PR, and AR expression (*p*: 0.008, *p*: 0.010, and *p*: 0.008, respectively) (Table 5). Multivariate analysis showed that histological type and AR expression are independent prognostic factors (OR: 0.36, 95% CI 0.18–0.75, *p*: 0.006; OR: 0.99, 95% CI 0.98–1.00, *p*: 0.050, respectively) (Table 5).

The Kaplan-Meier curve for OS showed no differences among BC HER2+ subtypes (Figure 2A). However, patients with G3 tumors had a worse survival (*p*: 0.004; Figure 2D). The best survival rate was found for those at stage I (*p* < 0.0001; Figure 2C). A significant OS reduction was associated with increasing levels of lymph node ratio (*p* < 0.0001; Figure 2B) and PR expression < 10% (*p*: 0.040; Figure 2E).

## 4. Discussion

BC HER2+ is an aggressive subtype including heterogeneous tumors with variable prognosis and treatment response to HER2-targeted therapies. On this matter, Staaf et al. identified three genetic BC HER2+ subtypes showing distinct clinical outcomes according to molecular profiling analysis [39]. The same group of researchers strengthened the molecular and biological complexity of BC HER2+ by showing the presence of high-level amplifications at multiple sites that also involved the HER2-amplicon at 17q12-q21 [39].

AR is a steroid hormone receptor frequently expressed in BC, including the ER-negative subtypes for which it could represent a complementary target for therapy, though the clinical significance and functional role of AR has not been outlined in BC yet [40,41]. The AR signaling pathway differs according to molecular breast cancer subtypes. It is known that in non-luminal HER2+ AR+ BCs, AR starts the WNT/β-catenin activation, stimulating HER3 gene transcription, subsequently the heterodimers between HER3 and HER2 support cell proliferation [42]. Nevertheless, the AR signaling pathway in luminal HER2+ AR+ BCs is unclear, and the differences between AR signaling pathways in HER2+ BCs subtypes are not fully understood.

AR expression is higher than ER and PR in BC [16,43]. We found a high prevalence (91.5%) of AR expression in BC HER2+, with minimal differences between luminal and non-luminal tumors; the absence or low level of AR expression being detected in a minority of tumors. These results are consistent with previous reports, which did not show significant differences between HER2-positive/ER-positive and HER2-positive/ER-negative tumors [29,34,44,45,46]. The higher percentage of AR expression in our cohort could depend on the definition of the AR expression, or on methods such as the use of the complete core section, population, and cut-off value.

Furthermore, our results showed that AR+ tumors are associated with older age at diagnosis and favorable clinical and pathological features, as well as lower histologic grade (G2), pT1 and prognostic stage (I), suggesting a prognostic value of AR in BC HER2+ subtypes. These findings are supported by previous studies showing the association of AR expression with lower grade, smaller size, more frequently tubule formation, and less pleomorphism and mitotic counts, although they evaluated ER-negative and ER-positive subgroups [17,47]. Besides, AR positivity was associated with high ER and PR expression in our cohort. Recently, Cruz-Tapias et al. evaluated the association of AR gene expression in accordance with intrinsic BC subtypes by meta-analysis of extensive microarray transcriptomic datasets. AR overexpression was prevalently observed in patients affected by less aggressive intrinsic molecular subtypes expressing either ER or PR and having a lower histological grade, such as Luminal A and B compared to Basal-like subtype. High AR mRNA levels can be defined as a prognostic biomarker for the detection of the less aggressive BC subtypes [48]. In the current study, the distinction of BC HER2+ in luminal and non-luminal showed that the lobular histotype is prevalent in luminal AR+ and AR− tumors compared to non-luminal subtypes. The apocrine phenotype can be found in non-luminal HER2 + AR+ tumors, whereas AR negativity increases in the mucinous phenotype in the non-luminal subtype. These data consent to distinguish a different phenotype between luminal and non-luminal BC HER2 + AR+ subtypes, adding more detail to Park’s results [17].

Moreover, the luminal BC HER2+ AR+ was associated with lower histological grade (G2) and lower tumor size, higher PR expression, and a lower HER2 intensity of expression (2+). Also, non-luminal tumors AR+ showed smaller size and prognostic stage (I) but higher grade (G3) and higher HER2 intensity of expression (3+). These results agree with the observations of other studies that have detected HER2 overexpression in G3-AR+ carcinomas [16,29,49]. These findings could suggest a different progression of luminal and non-luminal tumors both expressing AR, and how AR−related molecular mechanisms of BC HER2+ AR+ could differ depending on ER and PR expression.

Traditionally, the BC HER2+ subtype is distinguished in luminal HER2+ and non-luminal HER2+. In our study, AR positivity was detected in 91.5% of all tumors analyzed. Moreover, AR is expressed in 84.2% of BC HER2+, in the absence of other hormonal receptors, namely ER and PR. From a clinical point of view, tumors with AR expression show a slightly better clinical outcome. These findings suggest that the immunophenotype classification is not completely exhaustive; like TNBC [50], the presence of AR could confer a luminal phenotype. The concurrent presence with the estrogen receptor could make the androgen receptor a favorable prognostic marker also in the BC HER2+ subtype.

The role of androgens and AR might vary depending on cancer cell types and/or on the level of expression of other steroid hormone receptors. In ER-positive tumors, AR has an anti-proliferative effect by antagonizing ER, by binding to a subset of estrogen response elements (EREs); it can prevent the activation of target genes which mediate the stimulatory effects of 17-beta-estradiol on breast cancer cells [51,52]. Recently, Hickey et al. demonstrated that AR performs a tumor suppressor role in the ER-positive BC subtype, and the AR activation induces potent antitumor activity in multiple disease contexts, including resistance to endocrine therapy and CDK4/6 inhibitors. These data reinforce AR agonism as the optimal AR−directed treatment strategy, showing a rational therapeutic opportunity for this tumor subtype.

However, androgens may have a proliferative effect through AR in ER- AR+ tumor cells. In HER2+ tumors, AR triggers the WNT/β-catenin pathway causing HER3 upregulation; through HER2/HER3 heterodimers, it can cause the activation of the PI3K/AKT pathway and may lead to cell proliferation through MYC [53,54]. Furthermore, in non-luminal BC HER2+, AR induces HER2 expression, which in turn leads to ERK activation, which requires HER2 and AR activity. These findings suggest that HER2 is an upstream connector between the AR and ERK signaling pathways. Another feature of this feedback loop is an ERK-mediated regulation of AR [55]. Previous findings should have potential clinical relevance; considering the He’s study, the inhibition of AR with enzalutamide or shRNAs decreases the growth of HER2+ BC cells in vitro and in vivo, having a sensitivity similar to the trastuzumab. Interestingly, the inhibition of AR diminished the phosphorylation of HER2 and the activation of AKT and ERK without involving HER2 and HER3 protein expression levels. These findings indicate a new role of AR in HER2 signaling, and anti-AR target therapy may be beneficial in HER2+ BC patients that are unresponsive or that develop resistance to anti-HER2 therapies [56].

The prognostic role of AR in BC ER-negative and PR-negative has been poorly evaluated, in comparison with BC ER-positive PR-positive variants [57]. Our study, which was focused on luminal and non-luminal BC HER2+ showed that histological type and AR expression are good independent prognostic factors for OS in the HER2+ BC patients. Furthermore, it was found that a poorer OS in BC HER2-positive patients with higher histological grade (G3), higher level of lymph node ratio and PR expression <10%. Wang et al. demonstrated the association between BC HER2+ expressing AR and longer progression-free survival, increased five years OS rate, and the efficacy of trastuzumab therapy [34]. Akashi et al. found that AR expression was associated with the significant effectiveness of neoadjuvant chemotherapy and prognosis in HER2+ tumors [46]. Our study does have some limitations, that are primarily focused on its retrospective strategy. Hence, some information on clinical follow-up data were not completely included in the medical records. Additionally, the analysis should be expanded to more patients in the coming years to consolidate and reproduce our results.

## 5. Conclusions

AR is frequently expressed in BC HER2+ subtypes. Our results showed that AR expression has a prognostic value in BC HER2+ subtypes, with better clinical outcomes. A better prognosis was highlighted in luminal HER2+ AR+ tumors subtype compared to the non-luminal HER2+ AR+ subtype, based on clinico-pathological data. AR expression should be assessed to evaluate the prognosis of BC HER2+ subtypes. Moreover, the understanding of the complex interactions between AR and the HER2 signaling pathway could pave the way to the use of AR as a therapeutic target in BC HER2+ subtypes.

## Figures and Tables

**Figure 1 biomedicines-10-00164-f001:**
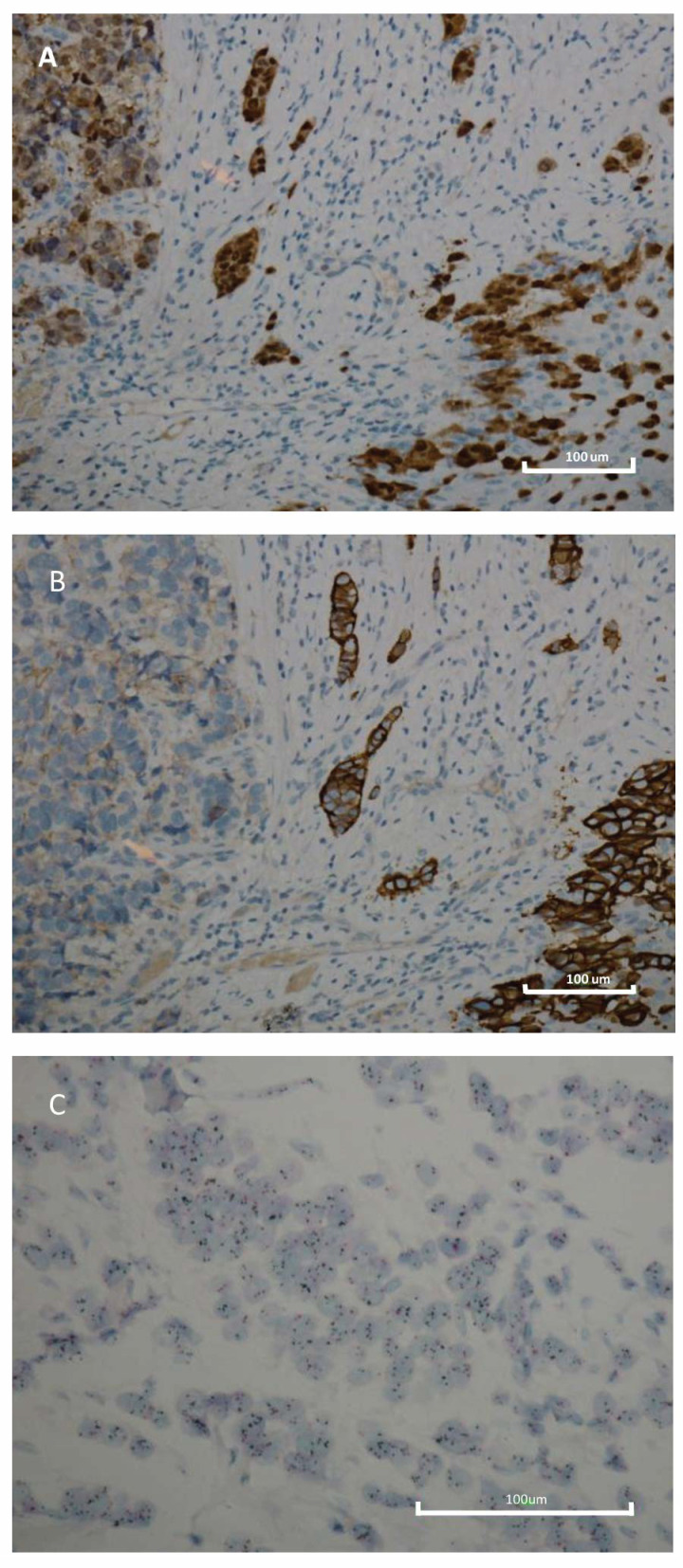
Immunohistochemistry and silver in situ hybridization features of breast cancer. (**A**) Immunohistochemistry for AR displaying diffuse and intense nuclear immunoreactivity (original magnification 200×); (**B**) Immunohistochemistry for HER2 displaying diffuse and intense membranous immunoreactivity (original magnification 200×); (**C**) Determination of HER2 gene status using the Dual SISH kit (Ventana) of a breast carcinoma with HER2 gene amplification; HER2 (black) and Chr17 (red). Scale bar: 100 μm.

**Figure 2 biomedicines-10-00164-f002:**
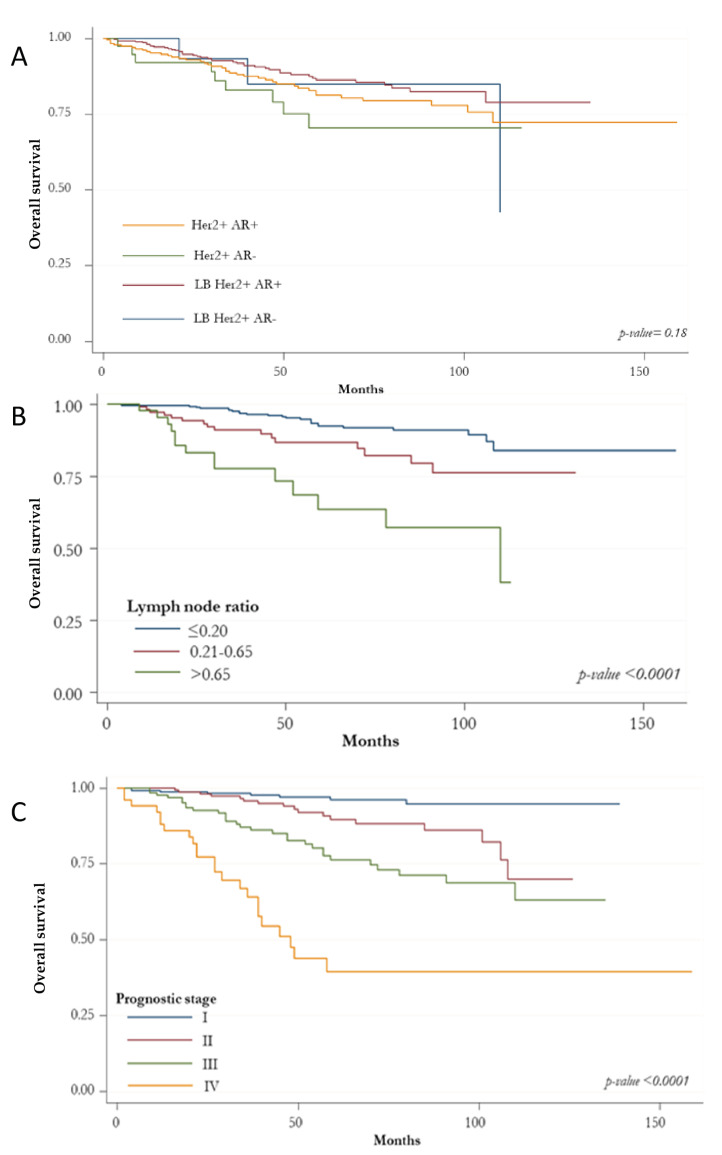
Kaplan-Meir curves for overall survival. (**A**) overall survival according to HER2-positive subtypes based on the state of the hormonal receptors (ER, PR and AR); (**B**) overall survival according to histologic grade; (**C**) overall survival according to prognostic stage; (**D**) overall survival according to lymph node ratio; (**E**) overall survival according to absence/presence of immunostaining for PR.

**Table 1 biomedicines-10-00164-t001:** HER2-positive BC subtypes proportion and immunohistochemistry positivity frequency.

IHC BC Subtypes	Number	%	HER2 IHC Intensity	
			2+	3+
Luminal B HER2 + AR+	396	57.0	126	270
Luminal B HER2 + AR−	18	2.6	6	12
HER2 + AR+	241	34.7	22	219
HER2 + AR−	40	5.7	5	35

IHC: Immunohistochemistry.

**Table 2 biomedicines-10-00164-t002:** Clinico-pathological features of HER2-positive breast cancer.

Variables		*n* = 695
Median (IQR) age, years		58 (49–68)
Age (year), *n* (%)	<58	332 (47.8)
	≥58	363 (52.2)
Site, *n* (%)	Left	396 (57.2)
	Right	296 (42.8)
Histologic type, *n* (%)	Ductal	590 (85.0)
	Apocrine	39 (5.6)
	Lobular	30 (4.3)
	Micropapillary/papillary	15 (2.2)
	Mixed (ductal + lobular)	13 (1.9)
	Mucinous	7 (1.0)
Histologic grade, *n* (%)	G2	112 (16.3)
	G3	575 (83.7)
Pathological tumor size, *n* (%)	pT0 *	53 (8.5)
	pT1	203 (32.6)
	pT2	288 (46.2)
	pT3	27 (4.3)
	pT4	52 (8.4)
Pathological lymph node status, *n* (%)	pN0	321 (55.9)
	pN1	134 (23.3)
	pN2	73 (12.7)
	pN3	46 (8.0)
Lymph node ratio, *n* (%)	≤0.20	400 (72.3)
	0.21–0.65	106 (19.2)
	>0.65	47 (8.5)
Prognostic stage, *n* (%)	I	249 (40.8)
	II	174 (28.5)
	III	134 (21.9)
	IV	54 (8.8)
Metastasis, *n* (%)		54 (7.8)
Proliferation index (Ki-67), *n* (%)	<14%	9 (1.3)
	≥14%	686 (98.7)
PR expression, *n* (%)	<10%	455 (65.5)
	≥10%	240 (34.5)
ER expression, *n* (%)	<10%	281 (40.4)
	≥10%	414 (59.6)
AR expression, *n* (%)	<10%	59 (8.5)
	≥10%	636 (91.5)
HER2 expression, *n* (%)	2+	159 (22.9)
	3+	536 (77.1)
Mortality, *n* (%)		99 (14.2)

IQR: Interquartile range, *n*: Number. * pT0 refers to pathological stage post neoadjuvant therapy (NAT).

**Table 3 biomedicines-10-00164-t003:** Clinico-pathological data of 695 patients with HER2-positive breast cancer based on AR expression.

		AR− (*n* = 58)	AR+ (*n* = 637)	*p*-Value
Median (IQR) age, years		55 (44–68)	59 (49–68)	0.045
Age (year), *n* (%)	<50	23 (39.7)	161 (25.3)	0.020
	≥50	35 (60.3)	476 (74.7)	
Site, *n* (%)	Left	32 (55.2)	364 (57.4)	0.740
	Right	26 (44.8)	270 (42.3)	
Histologic type, *n* (%)	CDI	52 (89.7)	538 (84.6)	0.770
	CLI	1 (1.7)	29 (4.6)	
	CDI + CLI	0 (0.0)	13 (2.0)	
	Apocrine	3 (5.2)	36 (5.7)	
	Micropapillary + papillary	1 (1.7)	14 (2.2)	
	Mucinous	1 (1.7)	6 (0.9)	
Histologic grade, *n* (%)	G2	3 (5.2)	109 (17.3)	0.020
	G3	55 (94.8)	520 (82.7)	
Pathological tumor size, *n* (%)	pT0 *	4 (7.6)	49 (8.6)	0.020
	pT1	11 (20.8)	192 (33.7)	
	pT2	25 (47.2)	263 (46.2)	
	pT3	7 (13.2)	20 (3.5)	
	pT4	6 (11.3)	45 (7.9)	
Pathological lymph node status, *n* (%)	pN0	20 (46.5)	301 (56.7)	0.430
	pN1	14 (32.6)	120 (22.6)	
	pN2	5 (11.6)	68 (12.8)	
	pN3	4 (9.3)	42 (7.9)	
Lymph node ratio, *n* (%)	≤0.20	26 (65.0)	374 (72.9)	0.470
	0.21–0.65	10 (25.0)	96 (18.7)	
	>0.65	4 (10.0)	43 (8.4)	
Prognostic stage, *n* (%)	I	8 (16.3)	241 (42.8)	0.001
	II	22 (44.9)	152 (27.0)	
	III	13 (26.5)	122 (21.7)	
	IV	6 (12.2)	48 (8.5)	
Metastasis, *n* (%)		6 (10.3)	48 (7.5)	0.440
Proliferation index (Ki-67), *n* (%)	<14%	0 (0.0)	9 (1.4)	1.000
	≥14%	58 (100.0)	627 (98.6)	
ER expression, *n* (%)	<10%	40 (69.0)	241 (37.8)	<0.0001
	≥10%	18 (31.0)	396 (62.2)	
PR expression, *n* (%)	<10%	51 (87.9)	404 (63.4)	<0.0001
	≥10%	7 (12.1)	233 (36.6)	
AR expression, *n* (%)	<10%	57 (98.3)	2 (0.3)	<0.0001
	≥10%	1 (1.7)	635 (99.7)	
HER2 expression, *n* (%)	2+	11 (19.0)	148 (23.2)	0.680
	3+	48 (81.0)	489 (76.8)	
Mortality, *n* (%)		12 (20.7)	87 (13.7)	0.140

IQR: Interquartile range, *n*: Number. * pT0 refers to pathological stage post NAT.

**Table 4 biomedicines-10-00164-t004:** Clinico-pathological data of 695 patients with HER2-positive breast cancer based on IHC subtypes.

		LB HER2+ AR− (*n* = 18)	LB HER2+ AR+ (*n* = 396)	HER2+ AR− (*n* = 40)	HER2+ AR+ (*n* = 241)	*p*-Value
Median (IQR) age, years		53.5 (41–62)	59 (48–69)	55 (45–70)	59 (50–68)	0.160
Age (year), *n* (%)	<50	8 (44.4)	110 (27.8)	15 (37.5)	51 (21.2)	0.030
	≥50	10 (55.6)	286 (72.2)	25 (62.5)	190 (78.8)	
Site, *n* (%)	Left	11 (61.1)	230 (58.4)	21 (52.5)	134 (55.8)	0.870
	Right	7 (38.9)	164 (41.6)	19 (47.5)	106 (44.2)	
Histologic type, *n* (%)	CDI	16 (88.9)	338 (85.4)	36 (90.0)	200 (83.3)	<0.0001
	CLI	1 (5.6)	25 (6.3)	0 (0.0)	4 (1.7)	
	CDI + CLI	0 (0.0)	10 (2.5)	0 (0.0)	3 (1.3)	
	Apocrine	1 (5.6)	8 (2.0)	2 (5.0)	28 (11.7)	
	Micropapillary + papillary	0 (0.0)	11 (2.7)	1 (2.5)	3 (1.3)	
	Mucinous	0 (0.0)	4 (1.0)	1 (2.5)	2 (0.8)	
Histologic grade, *n* (%)	G2	1 (5.6)	100 (25.5)	2 (5.0)	9 (3.8)	<0.0001
	G3	17 (94.4)	292 (74.5)	38 (95.0)	228 (96.2)	
Pathological tumor size, *n* (%)	pT0 *	2 (12.5)	20 (5.7)	2 (5.4)	29 (13.2)	0.007
	pT1	4 (25.0)	127 (36.3)	7 (18.9)	65 (29.6)	
	pT2	6 (37.5)	162 (46.3)	19 (51.4)	101 (45.9)	
	pT3	2 (12.5)	15 (4.3)	5 (13.5)	5 (2.3)	
	pT4	2 (12.5)	26 (7.4)	4 (10.8)	20 (9.1)	
Pathological lymph node status, *n* (%)	pN0	6 (40.0)	182 (55.8)	14 (50.0)	119 (58.1)	0.570
	pN1	4 (26.7)	75 (23.0)	10 (35.7)	45 (22.0)	
	pN2	3 (20.0)	46 (14.1)	2 (7.1)	22 (10.7)	
	pN3	2 (13.3)	23 (7.1)	2 (7.1)	19 (9.3)	
Lymph node ratio, *n* (%)	≤0.20	7 (50.0)	227 (70.9)	19 (73.1)	147 (76.2)	0.300
	0.21–0.65	6 (42.9)	64 (20.0)	4 (15.4)	32 (16.6)	
	>0.65	1 (7.1)	29 (9.1)	3 (11.5)	14 (7.3)	
Prognostic stage, *n* (%)	I	6 (35.3)	193 (55.5)	2 (6.3)	48 (22.3)	<0.0001
	II	4 (23.5)	64 (18.4)	18 (56.3)	88 (40.9)	
	III	4 (23.5)	60 (17.2)	9 (28.1)	62 (28.8)	
	IV	3 (17.7)	31 (8.9)	3 (9.4)	17 (7.9)	
Metastasis, *n* (%)		3 (16.7)	32 (8.1)	3 (7.5)	16 (6.6)	0.400
Proliferation index (Ki-67), *n* (%)	<20%	1 (5.6)	30 (7.6)	0 (0.0)	11 (5.0)	0.180
	≥20%	17 (94.4)	366 (92.4)	40 (100.0)	229 (95.0)	
	<14%	0 (0.0)	6 (1.5)	0 (0.0)	3 (1.2)	1.000
	≥14%	18 (100.0)	390 (98.5)	40 (100.0)	238 (98.8)	
ER expression, *n* (%)	<10%	0 (0.0)	0 (0.0)	40 (100.0)	241 (100.0)	<0.0001
	≥10%	18 (100.0)	396 (100.0)	0 (0.0)	0 (0.0)	
PR expression, *n* (%)	<10%	11 (61.1)	166 (41.9)	40 (100.0)	238 (98.8)	<0.0001
	≥10%	7 (38.9)	230 (58.1)	0 (0.0)	3 (1.2)	
AR expression, *n* (%)	<10%	17 (94.4)	0 (0.0)	40 (100.0)	2 (0.8)	<0.0001
	≥10%	1 (5.6)	396 (100.0)	0 (0.0)	239 (99.2)	
HER2 expression, *n* (%)	2+	6 (33.3)	126 (31.8)	5 (12.5)	22 9.1)	<0.0001
	3+	12 (66.7)	270 (68.1)	35 (87.5)	219 (90.9)	
Mortality, *n* (%)		3 (16.7)	45 (11.4)	9 (22.5)	42 (17.4)	0.060

IHC: Immunohistochemistry, IQR: Interquartile range, *n*: Number. * pT0 refers to pathological stage post NAT.

**Table 5 biomedicines-10-00164-t005:** Univariate and multivariate analysis for overall survival in HER2-positive breast cancer.

		Univariate Analysis		Multivariate Analysis	
		OR (95% CI)	*p*-Value	OR (95% CI)	*p*-Value
LB HER2 + AR−		1.21 (0.34–4.26)	0.770	-	-
LB HER2 + AR+		0.58 (0.38–0.89)	0.010	4.17 (0.29–59.39)	0.290
HER2 + AR−		1.82 (0.84–3.96)	0.130	-	-
HER2 + AR+		1.47 (0.95–2.27)	0.080	-	-
Age, years		1.03 (1.01–1.05)	<0.0001	1.02 (0.99–1.04)	0.200
Age ≥ 50 years		1.50 (0.89–2.53)	0.130	-	-
Histologic type, CDI VS. others		0.49 (0.29–0.81)	0.006	0.36 (0.18–0.75)	0.006
Histologic grade G3 VS. G2		4.08 (1.62–10.27)	0.003	2.21 (0.68–7.18)	0.190
Tumor size, from pT0 to pT4		2.49 (1.96–3.17)	<0.0001	-	-
Pathological tumor size	pT0 *	Ref.	-	-	
	pT1	1.05 (0.22–5.08)	0.960	1.31 (0.24–7.11)	0.750
	pT2	3.99 (0.93–17.07)	0.060	2.51 (0.53–11.87)	0.240
	pT3	4.44 (0.76–25.97)	0.100	2.56 (0.34–19.48)	0.360
	pT4	23.61 (5.20–107.30)	<0.0001	5.25 (0.87–31.47)	0.070
Pathological lymph node status, from pN0 to pN3		2.12 (1.65–2.71)	<0.0001	-	-
Pathological lymph node status	pN0	Ref.	Ref.	-	-
	pN1	2.40 (1.15–5.01)	0.020	1.20 (0.47–3.11)	0.700
	pN2	4.13 (1.89–9.03)	<0.0001	2.11 (0.49–9.15)	0.320
	pN3	10.17 (4.62–22.36)	<0.0001	3.17 (0.57–17.69)	0.190
Lymph node ratio		8.47 (3.76–19.10)	<0.0001	-	-
Lymph node ratio	≤0.20	Ref.	Ref.	-	-
	0.21–0.65	2.67 (1.37–5.20)	0.004	1.06 (0.35–3.21)	0.910
	>0.65	6.36 (3.02–13.40)	<0.0001	1.60 (0.45–5.88)	0.480
Prognostic stage, from I to IV		2.89 (2.22–3.76)	<0.0001	-	-
Prognostic stage	I	Ref.	Ref.	-	-
	II	3.26 (1.38–7.74)	0.007	1.43 (0.47–4.42)	0.530
	III	8.61 (3.82–19.40)	<0.0001	1.96 (0.45–8.52)	0.370
	IV	25.97 (10.72–62.89)	<0.0001	1.79 (0.23–14.14)	0.580
Metastasis		7.23 (4.02–12.99)	<0.0001	-	-
Proliferation index, %		1.02 (1.01–1.03)	0.002	1.02 (0.99–1.04)	0.150
Proliferation index (Ki-67) ≥14%		-	-	-	-
ER expression, %		0.99 (0.99–1.00)	0.008	-	-
ER expression ≥10%		0.59 (0.39–0.91)	0.020	0.17 (0.02–2.42)	0.190
PR expression, %		0.99 (0.99–1.00)	0.070	-	-
PR expression ≥10%		0.53 (0.32–0.87)	0.010	1.28 (0.47–3.44)	0.630
AR expression, %		0.99 (0.98–0.99)	0.008	0.99 (0.98–1.00)	0.050
AR expression ≥10%		0.62 (0.32–1.22)	0.170	-	-
HER2, %		1.00 (0.99–1.01)	0.480	-	-
HER2 expression 3 + VS. 2+		1.20 (0.71–2.03)	0.490	-	-

* pT0 refers to pathological stage post NAT.

## Data Availability

Not applicable.

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
