# Peer review of "Prognostic Role of Androgen Receptor Expression in HER2+ Breast Carcinoma Subtypes"

_biomedicines, 2022, doi:10.3390/biomedicines10010164_

Round 1

Reviewer 1 Report

This manuscript describes AR expression as a prognostic marker in breast cancer (BR) HER2+ subtypes. There are only minor concerns and suggestions. 

At the top of Table 1, n° should be replaced by “number” or another appropriate word. 

On page 4, N0 - N3 should be defined. 

In Table 2, IQR and pT0 – pT4 should be defined. 

On page 7 – “Luminal and non-luminal BC HER2+ AR+ showed a higher age at diagnosis. HER2+AR- and HER2+AR+ subtypes showed less frequently a lobular histotype.” – What do you compare to describe “higher” and “less”? 

In the Kaplan-Meier curve explanation on page 10, Fig2 A – D should refer to each explanation. 

In Conclusion, “A different prognosis” should be stated concretely. 

Please add the paper “Intrinsic Subtypes and Androgen Receptor Gene Expression in Primary Breast Cancer. A Meta-Analysis” Biology (Basel). 2021 Sep; 10(9): 834, Cruz-Tapias P et al., as a reference and discuss. 

Author Response

We are grateful for your consideration of this manuscript, and we thank the reviewers which have permitted us to improve the quality of our work with their suggestions. All the comments we received on this study have been considered and we have addressed all of them in our point-by-point reply below.

  1. At the top of Table 1, n° should be replaced by “number” or another appropriate word. 

According to the Reviewer’s comment, we changed “n°”  with “number”  in Table 1

  1. On page 4, N0 - N3 should be defined. 

According to the Reviewer’s comment, we specified that N0-N3 correspond to the pathological lymph node status (see Result section, pag. 4)

  1. In Table 2, IQR and pT0 – pT4 should be defined. 

According to the Reviewer’s comment, we defined IQR and pT0-pT4 in Table 2

  1. On page 7 – “Luminal and non-luminal BC HER2+ AR+ showed a higher age at diagnosis. HER2+AR- and HER2+AR+ subtypes showed less frequently a lobular histotype.” – What do you compare to describe “higher” and “less”? 

According to the Reviewer’s comment, we defined which BC subgroups are compared when describe the significant higher age at diagnosis and the frequency of  lobular histotype (see Result section, pag. 7)

  1. In the Kaplan-Meier curve explanation on page 10, Fig2 A – D should refer to each explanation. 

According to the Reviewer’s comment, we modified as suggested (see Result section, pag. 10).

  1. In Conclusion, “A different prognosis” should be stated concretely. 

According to the Reviewer’s comment, we modified as suggested (see Conclusion section pag.14)

  1. Please add the paper “Intrinsic Subtypes and Androgen Receptor Gene Expression in Primary Breast Cancer. A Meta-Analysis” Biology (Basel). 2021 Sep; 10(9): 834, Cruz-Tapias P et al., as a reference and discuss. 

According to the Reviewer’s comment, we discussed the suggested manuscript (see Discussion section, pag.13)

Reviewer 2 Report

Authors present a work addressing: ‘Prognostic role of androgen receptor expression in HER2+ breast carcinoma subtypes’. The aim of the study was to evaluate androgen receptor (AR) expression in a large series of HER2+ BC, to establish its prognostic role and clinical-pathological relationship. The general conclusion demonstrates that a different progression of luminal and non-luminal tumors both expressing AR and allowed Authors to speculate that the molecular mechanisms of AR, involved in the biology of BC HER2+AR+, differ in relation to ER and PR expression. Moreover, AR expression may be a useful predictor of prognosis for overall survival (OS) in HER2+ BC subtypes. The findings suggest that AR expression evaluation in clinical practice could be utilized in clinical oncology to establish different aggressiveness in BC HER2+ subtypes. The topic of the article is relevant for clinical practice. However, the paper presents a few major issues including:

  1. The abstract presents typical structure and did not require modifications.
  2. Please specify the period from which the authors selected study participants.
  3. Please provide inclusion and exclusion criteria.
  4. Expression of HER2 on 1 + is considered as negative thus authors should exclude those 2 patients. Moreover, scores of 2+ are considered as equivocal samples, which should be further analysed by fluorescence in situ hybridisation (FISH) confirmation, which detects gene amplification (Table 2, 3, 5).
  5. In table 2 authors should divide study group according to their median age- 58 years not by 50 years.
  6. The figures’ quality is very low (no 2 A-E). It should be improved by increasing of their size.
  7. The authors should provide limitations of the study.
  8. General: interesting, well conducted work.

Author Response

We are grateful for your consideration of this manuscript, and we thank the reviewers which have permitted us to improve the quality of our work with their suggestions. All the comments we received on this study have been considered and we have addressed all of them in our point-by-point reply below.

  1. Please specify the period from which the authors selected study participants.

Accordingly, with reviewer’s suggestion, we added the period of time in which the participants were selected (see Materials and Methods section pag. 2)

  1. Please provide inclusion and exclusion criteria.

Accordingly, with reviewer’s suggestion, we added the inclusion and exclusion criteria to select participants (see Materials and Methods section pag. 2-3)

  1. Expression of HER2 on 1 + is considered as negative thus authors should exclude those 2 patients. Moreover, scores of 2+ are considered as equivocal samples, which should be further analysed by fluorescence in situ hybridisation (FISH) confirmation, which detects gene amplification (Table 2, 3, 5).

Thank you for the comments. The two patients with HER2 IHC 1+ were evaluated incongruously (IHC 1+ and IHC 2+) by two pathologists and subjected to SISH, which resulted positive for gene amplification. Then we included the previous two patients together with patients with HER2 IHC 2+ and SISH positive.

Moreover, in Materials and Methods section pag. 3, we defined that “In cases of HER2 with equivocal immunohistochemical score (2+), we performed HER2 gene amplification by ultra-View SISH Detection Kit (Ventana Medical Systems)”. We added the phrase: “All patients with HER2 IHC 2+ included in the study are SISH positive” in Results section pag. 4 .

  1. In table 2 authors should divide study group according to their median age- 58 years not by 50 years.

Accordingly, with reviewer’s suggestion, we stratified the study group according to median age- 58 years in Table 2

  1. The figures’ quality is very low (no 2 A-E). It should be improved by increasing of their size.

Accordingly, with reviewer’s suggestion, we increased the size of Figure 2.

  1. The authors should provide limitations of the study.

Accordingly, with reviewer’s suggestion, we described limitations of the study (see Discussion section, pag.14)

Round 2

Reviewer 2 Report

The current version is much better than previous one. I suggest to publish this manuscript in current version.

Method section:

I am just wondering if this is the correct abbreviation SISH and it should not be ISH or FISH? Since according to manufacturer instruction you can see: The new VENTANA HER2 Dual ISH DNA Probe Cocktail assay is a fully automated, ready-to-use brightfield solution for determining HER2 gene status. VENTANA HER2 Dual ISH helps identify breast and gastric cancer patients eligible for treatment with HER2-targeted personalized therapies.

or previously it was used Antibodies VENTANA anti-HER2/neu (4B5) was applied for laboratory semiquantifica- tive detection of the antigen HER2/neu with the use of a VENTANA aperture for staining the IHC microscopic slide (Benchmark Ultra, Roche-Ventana). HER2 was considered positive with a level of (+++) and negative with a level of (–), (+). However, scores of (++) were taken as equivocal cases, which were further recommended for fluores- cence in situ hybridization (FISH) confirmation, which detects gene amplification. FISH was performed using a dual HER2/Cep17 probe.